# Sustainable Synthesis of Iron–Zinc Nanocomposites by *Azadirachta indica* Leaves Extract for RSM-Optimized Sono-Adsorptive Removal of Crystal Violet Dye

**DOI:** 10.3390/ma16031023

**Published:** 2023-01-23

**Authors:** Hajira Tahir, Muhammad Saad, Osama A. Attala, Waleed A. El-Saoud, Kamal A. Attia, Shaista Jabeen, Jahan Zeb

**Affiliations:** 1Department of Chemistry, University of Karachi, Karachi 75270, Pakistan; 2Department of Environmental and Health Research, The Custodian of the Holy Mosques Institute for Hajj and Umrah Research, Umm Al-Qura University, Makkah 21955, Saudi Arabia; 3Natural Hazards Research Unit, Department of Environmental and Health Research, Umm Al-Qura University, Makkah 21955, Saudi Arabia; 4Biology Department, Al-Jammoum University College, Umm-Alqura University, Makkah 24381, Saudi Arabia

**Keywords:** adsorption, desorption, nanocomposites, response surface methodology (RSM), isotherms, ultrasonication, wastewater treatment

## Abstract

Environmental pollution has exacerbated the availability of clean water to mankind. In this study, *Azadirachta indica* leaf extract was used for sustainable synthesis of Fe–Zn nanocomposites (IZNC). The instrumental techniques of Fourier transformed infrared (FTIR) spectroscopy, energy dispersive X-ray spectroscopy (EDS), and scanning electron microscopy (SEM) were used to determine the structural and chemical composition. The overall surface was mildly acidic in nature, as the pH_PZC_ was observed to be 6.00. The ultrasonicated adsorption experiments were designed by central composite design (CCD). The best responses, which proposed a contaminants removal of 80.39%, were assessed using the response surface methodology (RSM). By repeating experimental runs at the expected optimum operating parameters (OOP), the method was experimentally affirmed with the %mean error and %RSD_9_ being 2.695% and 1.648%, respectively. The interaction of CV dye and the nanocomposite showed tremendous adsorption efficiency towards crystal violet (CV) dye, as revealed by isotherm studies. Fitting kinetics and isotherm models were affirmed by root mean square error (RMSE), χ2, and a Pearson regression coefficient. Thermodynamic studies proved spontaneity of the CV dye adsorption over the nanocomposites. The values for ΔG^o^, ΔH^o^, and ΔS^o^ were observed to be −1.089 kJ/mol, 28.59 kJ/mol, and −3.546 kJ/mol, respectively. Recovery of CV dye was carried out in a variety of media, including NaOH, NaCl, and CH_3_COOH. The maximum CV recovery was achieved in an acidic media. The robustness of adsorption was affirmed by the interference of various matrix ions, including KCl, LiCl, NaCl, and MgCl_2_, which did not significantly affect the adsorption process. The maximum adsorption capacity was obtained at a low concentration of LiCl. The results show that a green synthesis approach for nanocomposite synthesis might be an effective and economical way to remove organic contaminants from wastewater. Moreover, it is also effective for effluent treatment plants (ETP) for waste management purposes, in which it may be coupled with chlorine as a disinfectant to purify water that can be used for domestic and irrigation purposes.

## 1. Introduction

Nowadays, one of the most popular studies is green nanotechnology. The synthesis of nanoparticles using a green approach to reduce the use of toxic substances is an environmentally friendly approach [1]. Consequently, scientists are becoming interested in a green approach. The advantage of this strategy is its ability to modify particle size and stability by utilizing a rapid synthesis approach. A number of industries utilize nanoparticles because of their exceptional characteristics [2,3,4,5,6].

Regarding the green source of the synthesis of nanoparticles, *Azadirachta indica*, also locally known as “neem”, has been used to synthesize nanoparticles compared to other environmentally friendly sources. It is well known that the extract of the *A. indica* plant leaves contains enzymes and phytochemicals that are responsible for transforming forerunner metals (ions) into their nanoparticles [7]. For the production of nanoparticles of zinc oxide (ZnO), manganese oxide (Mn_3_O_4_), copper oxide (CuO), silver (Ag), platinum (Pt), and other elements, *A. indica* leaf extract has been used by several researchers [8,9,10,11,12]. Because of their higher surface area, nanoparticles are helpful for treating wastewater [13].

The nanoparticles of magnetic oxides and zinc oxide are appreciably used because of their extraordinary properties. Moreover, these are integrated with activated carbon, carbon nanotubes and polymers, which are applied for various purification processes [14].

Numerous approaches, such as biological, physical, and chemical treatment procedures, have been researched for the treatment of water. Ultrasonicated adsorptions processes have become one of the most efficient treatment methods. The ultrasonication approach is capable of removing and adsorbing various kinds of biologically resistant pollutants using low-cost methods. As a result, they improve the surface area and mass transfer proportion between the adsorbate and the adsorbent [15]. The utilization of Artocarpus odoratissimus (also locally known as Tarap) leaves (TL) for the removal of malachite green dye on an adsorbent is reported and the adsorption capacity was found to be 254.93 mg g^−1^ [16]. In this study, the adsorption of methylene blue (MB) dye using an aquatic plant, Azolla pinnata (AP), was modelled using several various supervised machine learning (ML) algorithms, aiming to accurately predict the adsorption capacity under various experimental conditions [17].

Fast removal of dyes and other contaminants is achieved by combining ultrasonic adsorption capacities with the unique features of nanoscale materials. Environmental pollution, which has risen sharply in recent decades, is one of the most significant current issues. The annual production of about 10,000 distinct varieties of colorants is thought to be around 700,000 tons. It is anticipated that a considerable proportion of these dyestuffs will pollute the environment by making their way into water bodies through wastewater.

The dyestuff poses several risks to people. For instance, the organic pollutant crystal violet can impair breathing problems and produces nausea, hemolysis, and hypertension. Decontamination of wastewater from toxic organic contaminants like crystal violet is therefore necessary to reduce environmental hazards [18,19,20,21].

Unwanted chemical and toxins can easily be removed from an aqueous medium with assistance from ultrasonic waves. Therefore, hybridization of ultrasonication with adsorption has been growing for treating wastewater containing various hazardous pollutants [22,23,24]. Ultrasonic irradiation facilitates adsorption due to acoustic cavitation. It relates to the initiation, propagation, and termination of micrometrical bubbles that are produced because of a pressure wave through the liquid [25]. In addition to cavitation, mass transfer is also increased due to modulation of the interaction between the adsorbate and adsorbent via convection without any considerable change in adsorption equilibrium [26,27,28].

The response surface approach is a prevalent way to optimize an adsorption process. It is used in conjunction with the central composite design (CCD), which allows for the performance of only selected runs and to optimize an experiment with the fewest possible experiments. In addition to reducing data duplication, it also makes it possible to evaluate intervariable impacts, which is not achievable with a one-variable approach. CCD is a sophisticated method for both experimental design and data redundancy avoidance [29]. Other statistical methods, such as Pareto analysis and analysis of variance (ANOVA), are also employed to evaluate the significance of the data.

Fe–Zn nanocomposites synthesized by *A. indica* leaves extract were used as an adsorbent in the current studies for the statistical adsorptive decontamination of crystal violet dye. Their thermodynamics, kinetics, and desorption properties were also investigated.

## 2. Materials and Methods

### 2.1. Chemicals

All the chemicals used in the current research were of analytical grade. FeSO_4_·7H_2_O, ZnSO_4_·6H_2_O, NaOH, LiCl, MgCl_2_, and KCl were purchased from Merck with a purity >99%. HCl and CH_3_COOH were purchased from Sigma Aldrich with a purity >97%.

Throughout the present research, chemicals of analytical grade were used. Table 1 provides the specifics on the raw materials employed in the current experiments.

The crystal violet dye was purchased from Sigma Aldrich. It has a molecular formula of C_24_H_28_N_3_Cl, and the corresponding molar mass is 408 g/mol. It is slightly soluble in water, but readily soluble in ethanol. Its λ_max_ is at 632 nm.

### 2.2. Preparation of Adsorbate (CV Dye) Standards and Test Solutions by Standard Addition Method

The standard adsorbate solutions were prepared by making a stock solution by dissolving 0.1 g of CV dye in 25 mL of ethanol. After that, the solution was transferred into a 500 mL volumetric flask. The volume of the flask was chosen as 500 mL to make a stock solution of 250 mg L^−1^. The dilutions from the stock were prepared by the stock addition method.

### 2.3. Sustainable Fe–Zn Nanocomposites Synthesis

The IZNC were synthesized by a green method. *A. indica* leaves were collected and subjected to repeated rinsing with tap water and then deionized distilled water. After being chopped into small pieces, the leaves were air-dried. Leaves extract of 25% (*w*/*v*) strength was prepared and heated at 80 °C for one hour in a hot air oven. The content was cooled to room temperature and filtered for the removal of unwanted residue. This extract of *A. indica* leaves was transferred into a mixture of 0.1 M ZnSO_4_·7H_2_O and 0.1 M FeSO_4_·7H_2_O at a ratio of 2:1, and the mixture was stirred for 4 h.

After adding the leaf extract, a color change from yellow to brown and then black was noticed in the mixture, indicating that the metallic precursor ions had been converted to their nanoparticle equivalents. The resulting nanoparticles were first filtered by passing through a filtration bed, and after that they were washed with deionized distilled water in a separate unit repeatedly, then dried at 60 °C. They were stored in an airtight desiccator for future use [30].

### 2.4. Nanocomposites Characterization

The synthesized nanocomposites were characterized by Fourier transformed infrared spectroscopy (FTIR). Model NICOLET 67000 was operated. The energy dispersive X-ray spectroscopy (EDS) and scanning electron microscopy (SEM), JEOLjapan product model# JSM6380A was used.

### 2.5. pH at Point of Zero Charge (pH_PZC_)

The pH_PZC_ surface characterization was used to determine surface neutrality. The overall surface charge of a species becomes negative above the pH_PZC_ and positive below the pH_PZC_, respectively.

The pH drift method was adopted to determine the pH_PZC_ of the IZNC. This method involved adjusting the pH of 50 mL of a NaCl solution to maintain 2.0–12.0 pH values (6 levels) by adding an acid or a base. After that, 10 mg of the nano-adsorbent was added into the solution. Subsequent to that, all the sets were kept at room temperature for 24 and 48 h. The surface neutrality was determined by a graphical method after 24 and 48 h by choosing the common point on both curves at the abscissa [15].

### 2.6. Ultrasonic Waves-Assisted Adsorption Experiments

The adsorbate (CV) dye, in accordance with CCD, was prepared at various concentrations. The pH of the test solutions was adjusted to set the CCD requirements before the specified amount of the nanocomposite was added. In an ultrasonication process, the specified set of solutions were sonicated and filtered by an adsorbent bed. The absorbance at the maximum concentration of CV dye was measured using a UV-visible spectrophotometer before and after the adsorption procedure. Attenuation in the absorption beam yielded the % removal of the CV dye, as shown in Table 1 [14].

### 2.7. Measurement of Adsorption Effectiveness

A series of standard CV dye solutions were evaluated to make a calibration curve. The following equation was used to evaluate the removal efficacy.
(1)$$\%\;Removal=\left(\frac{C_{o}-C_{t}}{C_{o}}\right)\times 100\%$$
where C_o_ is the initial concentration (mg L^−1^) of CV and C_t_ is the concentration (mg L^−1^) after removal at different time intervals. The equation given below was used to calculate the adsorption effectiveness.
(2)$$q_{e}=\frac{\left(C_{o}-C_{e}\right)V}{W}$$
where C_e_ is the equilibrium concentrations of CV dye (mg L^−1^). Additionally, (w) is the mass of the nanocomposites (g), and (V) is the volume of the solution in (dm^3^).

### 2.8. Central Composite Design (CCD)

Central composite design is frequently used as a base for a quadratic response surface model when determining the ideal operating parameters. By increasing the central points and decreasing the experimental error, the CCD accuracy is enhanced. The following equation calculates the RSM’s alpha value.
(3)$$\alpha =\sqrt[{4}]{2^{f}}$$

The total CCD experimental runs are calculated using the following relationship.
(4)$$N=2^{f}+2f+N_{o}$$

In the relationship above, the number of factors is denoted by f, the factorial points are denoted by 2^f^, and N_o_ shows central points.

In this study, 4 different factors of the CCD were utilized with 6 central points and 5 levels. An alpha with a 2.0 value was used for the design. As per CCD, 26 runs were completed in all. In the CCD-based RSM analysis, the ultrasonication time was denoted by ‘A’ with a range of 2 min to 10 min to study the change in adsorption with respect to time. The solution pH was indicated by ‘B’ and the study was performed within the pH values of 3.0–9.0 under acidic, neutral, and basic conditions. The amount of adsorbent was represented by ‘C’ and the adsorption was studied within 5 mg to 25 mg mass of the adsorbent. The pollutant concentration was represented by ‘D’ and the adsorption study was performed within 5 mg/L to 25 mg/L concentration of the adsorbate, beyond which the Beer–Lambert law would have deviated [15]. Table 1 provides the CCD and the accompanying replies.

The experimental runs were completed by using the quadratic polynomial equation, and the results were linked with four different factors.
(5)$$y=\beta_{o}+\overset{5}{\underset{i=1}{\sum }}\beta_{i}x_{i}+\overset{5}{\underset{i=1}{\sum }}\overset{5}{\underset{j=1}{\sum }}\beta_{ij}x_{i}x_{j}+\overset{5}{\underset{i=1}{\sum }}\beta_{ii}x_{i}^{2}+\varepsilon$$
where y is the expected (percentage removal), β_o_, β_i_, β_ii_, and β_ij_ are the model constant, linear, quadratic, and interaction coefficient, respectively, and X_i_ is the independent variable.

### 2.9. Response Surface Methodology (RSM)

The response surface approach and CCD are integrated in order to assess the critical variables and resolve multivariate equations to arrive at the optimal operating parameters. The first- or second-order polynomial equations are derived in accordance with the experimental findings and the consequent fitting of those findings. ANOVA makes it easier to assess all the experimental factors and their interactions based on their P and F-values. A 3D RSM graph was utilized to anticipate the optimal operating settings [15].

### 2.10. Adsorption Isotherms

The interactions between the adsorbent and adsorbate were evaluated by using several models of isotherms. The Dubinin–Radushkevich, Freundlich, Langmuir, and Temkin isotherm models were applied. The isotherm studies were carried out between 303–313 K temperatures. Various isotherm constants were evaluated and they were used to assess the interactions between the adsorbate and adsorbent [31].

#### 2.10.1. Langmuir Adsorption Isotherms

The Langmuir isotherm is the adsorption isotherm that presumes a chemical bond formation between the adsorbate and adsorbent’s surface that causes monolayered adsorption [14].
(6)$$\frac{C_{e}}{Q_{e}}=\frac{C_{e}}{Q_{o}}+\frac{1}{K_{L}Q_{o}}$$

In the equation above, C_e_ denotes the concentration of the adsorbate at equilibrium (mg L^−1^), C_o_ represents the initial concentration of the adsorbate (mg L^−1^), Q_o_ indicates the maximum adsorption capacity (mg g^−1^), and K_L_ represents the Langmuir constant related to adsorption energy (L mg^−1^).

#### 2.10.2. Freundlich Adsorption Isotherm

The idea of this model is that the presence of several functional groups on the adsorbent surface causes it to have a heterogeneous surface [15].
(7)$$Log\frac{x}{m}=Log\;K+\frac{1}{n}\log C_{e}$$
where K represents the Freundlich constant related to the adsorption coefficient (mg g^−1^).

#### 2.10.3. Dubinin–Radushkevich Adsorption Isotherm

This isotherm model assumes a pore-filling mechanism of the adsorption process. This is largely applicable to physical adsorption processes and takes into consideration the multilayer development based on van der Waals forces [32].
(8)$$Ln\;Q_{e}=Ln\;Q_{s}-\beta \varepsilon^{2}$$

In the equation above, Q_s_ is the maximum adsorption capacity mg g^−1^, β is a constant corresponding to adsorption energy ((mol^2^ kJ^−2^), and ε is the adsorption potential.

#### 2.10.4. Temkin Adsorption Isotherm

The idea of this model is that the energy of adsorption reduces linearly with the adsorbent surface coverage [32].
(9)$$Q_{e}=\frac{RT}{b_{T}}\ln A_{T}+\frac{RT}{b_{T}}\ln C_{e}$$
where A_T_ shows the binding constant that corresponds to the maximum binding energy (L mg^−1^) and b_T_ is associated with the adsorption heat (kJ mol^−1^).

#### 2.10.5. Harkins–Jura Model

This model takes into consideration the porous adsorption and multilayer formation on the heterogenous surface [32].
(10)$$\frac{1}{Q_{e}^{2}}=\frac{B_{H}}{A}-\frac{1}{A}\log C_{e}$$

In the equation above, B_H_ shows the Harkins–Jura model constant related to adsorption.

### 2.11. Kinetics Studies

The intraparticle diffusion model, Ho and McKay pseudo-second-order model, Lagergren pseudo-first-order model, and Elovich model were some of the kinetic models used in the adsorption kinetic research. The intercept and slope of the graphs (model equations) were used to calculate the values of the respective model constants [31].

#### 2.11.1. Pseudo-First-Order Kinetic Model

This is one of the most conventional models applied in studying the adsorption rate for the solid–liquid interface [33].
(11)$$\log(q_{e}-q_{t})=\log q_{e}-\frac{k_{1}t}{2.303}$$

#### 2.11.2. Pseudo-Second-Order Kinetic Model

This model is unarguably the most famous model for the kinetic modelling of adsorption data. This model depends on the initial concentration of the adsorbate [33].
(12)$$\frac{t}{q_{t}}=\frac{1}{k_{2}q_{e}^{2}}+\frac{t}{q_{e}}$$

#### 2.11.3. Intraparticle Diffusion Model

This isotherm model assumes the intraparticle diffusion, film diffusion, and pore-filling mechanism involved in the adsorption process. This takes into consideration the production of several layers based on van der Waals forces, which is primarily relevant to the development of physical adsorption processes [34].
(13)qt=Kidt12+C

#### 2.11.4. Elovich Model

The idea of the Elovich model is that the adsorption heat is reduced linearly along with the adsorbent’s surface area coverage [34].
(14)$$q_{t}=\frac{1}{b}\ln\left(ab\right)+\frac{1}{b}\ln t$$

In the equations above, k_1_, k_2_, k_id_, ‘a’, and ‘b’ denote the pseudo-first-order rate constant, the pseudo-second-order rate constant, the intraparticle diffusion coefficient, the rate of initial adsorption (mg g^−1^ min^−1^), and the desorption constant (g mg^−1^), respectively.

### 2.12. Error Analysis

The matching of various kinetics models and isotherms with the experimental data was confirmed by error analysis. In the present investigations, several model fits were analyzed using Chi-square (χ^2^), root mean square error (RMSE), and R^2^ values.

#### 2.12.1. Chi-Square (χ^2^) Test

To evaluate the differences between the experimental and model data, this statistical function is applied. The following equation is used to calculate the χ^2^ value for a model.
(15)$$\chi^{2}=\sum \frac{{\left(Q_{e,\;exp}-Q_{e,cal}\right)}^{2}}{Q_{e,cal}}$$
where Q_e,cal_ (mg g^−1^) is the adsorption capacity of the equilibrium model and Q_e,exp_ (mg g^−1^) is the adsorption capacity of the experiment. The size of the χ^2^ values is determined by how closely the experimental data resemble the model data. Higher χ^2^ values show poor fitting of the model and low χ^2^ values show nice fitting of the model [35].

#### 2.12.2. Root Mean Square Error (RMSE)

Another statistical tool to evaluate how well different models match the experimental data is a root mean square error. When evaluating RMSE for a model, apply the equation below.
(16)$$RMSE=\sqrt{\frac{1}{N-2}\overset{N}{\underset{i=1}{\sum }}{\left(Q_{e,exp}-Q_{e,cal}\right)}^{2}}$$

In the relationship above, the experimental adsorption capacity and model’s adsorption capacity have been represented as Q_e,exp_ (mg g^−1^) and Q_e,cal_ (mg g^−1^), respectively. Similar to the χ^2^ test, low values show good fitting, and high values show bad model fitting [36].

### 2.13. Thermodynamic Studies

Thermodynamics studies were used to evaluate the adsorption process’s spontaneity and viability. Studies on thermodynamics were conducted between 308 K and 318 K. The values of the thermodynamic constants were assessed using a van’t Hoff plot of the relationship between ln K_D_ and T^−1^ [15].

### 2.14. CV Dye Recovery

Studies on desorption for the recovery of the adsorbate using CH_3_COOH, NaCl, and NaOH were conducted on the exhausted adsorbent. Desorption tests were conducted for a duration of 2 to 10 min using the aforementioned species at media concentrations between 0.001 M and 0.100 M [37].

### 2.15. Interference Ions Effects

Natural wastewater systems contain diverse inorganic ionic species; hence, performance of the proposed RSM-optimized approach of adsorption was observed in the competition of diverse inorganic salts. By observing the adsorption of the adsorbate at OOP in the presence of various concentrations of LiCl, MgCl_2_, KCl, and NaCl, the impact of various salts on the adsorption process was assessed [38].

## 3. Results and Discussion

### 3.1. Characterization of Nanocomposites

Fourier transformed infrared spectroscopy (FTIR), energy dispersive X-ray spectroscopy (EDS), and scanning electron microscopy (SEM) were used to characterize the nanocomposites.

#### 3.1.1. Fourier Transformed Infrared Spectroscopy

FTIR analysis of the IZNC was used to gather data on the chemical bonds and their vibrations. Figure 1 displays the FTIR spectrum.

Figure 1 shows numerous peaks at various locations. The Zn–O bond’s vibration is thought to be responsible for the peak at 480.28 cm^−1^ [39]. There were several minor peaks at 534.28 cm^−1^ and 580.00 cm^−1^ that might have been caused by the vibrations of the Fe–O bonds [40]. The absorption maximum at 808.17 cm^−1^ is due to the bond vibration of aromatic C–H groups. According to the literature, the presence of aromatic ester or alkyl aryl ether in aqueous leaves extract of *A. indica* exhibits C–N bond vibrations and –C–O bond vibrations, which cause a number of additional peaks to arise at 1085.92 cm^−1^, 1209.37 cm^−1^, and 1273.02 cm^−1^ [41]. Due to the environment’s absorption of CO_2_ and the asymmetric and symmetric modes of vibration of –C=O, O–C=O, and O–C–O, the monodentate metal carbonate (m-CO_3_^2−^) appears to vibrate at 1355.96 cm^−1^, 1429.25 cm^−1^, 1469.76 cm^−1^, and 1533.41 cm^−1^ [42,43,44,45]. The peaks at 1620 cm^−1^, 1643 cm^−1^, and 2374.37 cm^−1^ are caused by the water molecules’ bending, as they are present on the surface of the synthetic material as a result of atmospheric adsorption [46,47]. The bond vibrations of the C–H groups are said to be responsible for the peaks at 2860.43 cm^−1^ and 2926.01 cm^−1^ [48]. The broad peaks at 3431.36 cm^−1^ and 3753.48 cm^−1^ are connected to the OH group bond vibrations on the surface of the produced nanocomposites [49].

#### 3.1.2. Scanning Electron Microscopy

Scanning electron microscopy was used to see the morphology of the surface of the IZNC. Figure 2 shows the SEM images of the synthesized nanoparticles.

Figure 2 demonstrates that the synthesized nanocomposites had a uniform size and shape. The bulk of them were found to be spherical. The average nanoparticle size, as shown in the SEM pictures, was 100 nm, which resulted in the enhanced adsorption efficacy of the nanoparticles [50].

#### 3.1.3. Energy Dispersive X-ray Spectroscopy

The elemental composition of the synthesized nanocomposites was ascertained by energy dispersive X-ray spectroscopy. Figure 3 displays the EDS spectra and the percentage composition of each element.

Iron, zinc, oxygen, and carbon were the main components of the nanocomposites, according to an elemental analysis performed by EDS. The use of green extract with a high carbon content in the production of the nanocomposite may be the cause of the carbon-related maxima in the spectrum. The atomic percentage of iron and zinc manifested that the synthesis of the iron nanoparticles was more facilitated by the *A. indica* extract as compared to the zinc nanoparticles [50].

#### 3.1.4. pH at Point of Zero Charge

The point of intersection of a plot of initial pH against ΔpH after 24 and 48 h was determined to be pH_PZC_. As seen in Figure 4, the curves crossed at pH 6.00 on abscissa, showing that the pH_PZC_ of the nanocomposites was at this p. This demonstrates that the nanocomposites have an overall positive superficial charge on the surface when pH < pH_PZC_ and an overall negative superficial charge when pH > pH_PZC_ [29,50].

### 3.2. Central Composite Design (CCD)

In this study, CCD was applied. Four factors, five levels, and six center points made up the CCD. These factors cover CV concentration, adsorbent quantity (g), time (min), and pH. There were 26 tests in all, and the results were utilized to create reaction surface graphs. The CCD is shown in Table 1 along with the observed responses.

**Table 1 materials-16-01023-t001:** Central composite design and the observed responses for the decontamination of CV from an aqueous medium.

S. No.	Time (min)	pH	Amount of Adsorbent (g)	CV Concentration (mg L^−1^)	% Removal
01	6	7	0.015	5.0	34.90
02	4	9	0.010	20	36.84
03	6	7	0.015	15	76.06
04	4	5	0.010	20	27.07
05	8	5	0.010	20	51.87
06	4	5	0.010	10	19.02
07	8	5	0.020	10	68.10
08	2	7	0.015	15	38.27
09	4	5	0.020	10	13.84
10	8	9	0.010	20	58.08
11	6	7	0.015	15	74.77
12	8	5	0.010	10	22.91
13	6	7	0.015	15	77.48
14	6	7	0.025	15	85.33
15	4	9	0.020	20	41.95
16	4	5	0.020	20	37.70
17	6	7	0.015	15	73.98
18	6	7	0.015	15	75.98
19	4	9	0.010	10	21.52
20	8	9	0.020	10	23.21
21	8	9	0.010	10	65.64
22	6	11	0.015	15	82.65
23	10	7	0.015	15	63.80
24	6	7	0.015	25	39.35
25	6	7	0.005	15	80.02
26	4	9	0.020	10	20.50
27	6	3	0.015	15	31.84
28	8	5	0.020	20	64.85
29	8	9	0.020	20	41.20
30	6	7	0.015	15	79.73

### 3.3. Analysis of Variances

To examine the significance of the applied model and the impact of each factor and combination, ANOVA must be used on the collected responses [14]. Table 2 provides the ANOVA results for the decontamination of dye (CV) by the IZNC.

The model’s *p*-value shows that it is significant. It is evident from the table that the most influential factor among all was time (min) for the sono-adsorptive removal of CV dye using the IZNC.

### 3.4. Pareto Analysis

The Pareto chart illustrates the effect made by each element and its combination. It is another good graphical method of demonstrating the relevance of the factors and their combinations for the model [15]. Figure 5 shows the Pareto chart visualization for the use of the nanocomposite (Fe–Zn–Al) for the adsorptive removal of dye (CV).

Figure 5 makes it evident that ultrasonication irradiation time (min) is the most important factor in the CV adsorption over IZNC having its Pareto coefficient of 9.52. It is followed by CV dye concentration (mg/L) that has a Pareto absolute coefficient of 4.74. Among all the interactions, DD was the most significant, followed by AA, BB, and BC, as indicated by their Pareto coefficients.

### 3.5. Normal Probability Plots

A statistical test to determine if the data points are inside the 95% confidence level is a normal probability plot. Figure 6 shows the normal probability plot that was created for the current research.

The data points are proportionally positioned around the central line, as shown in Figure 5. Since no data point was found outside of the 95% confidence level, this confirms that the CCD utilized in the method was appropriate [15].

### 3.6. Response Surface Methodology (RSM)

Response surface methodology is a widely used method for the process of optimization. The best response at the optimum operating parameters was obtained in the current studies using the quadratic RSM model. The results were utilized to create the model equation that is shown below.
% Removal CV = 76.33 + (9.520)*A + (4.384)*B + (0.7931)*C + (4.739)*D + (−2.672)*AB + (−0.6663)*AC + (−2.035)*AD + (−7.428)*BC + (−0.6528)*BD + (0.9555)*CD + (−9.429)*AA + (−7.877)*BB + (−1.518)*CC + (−12.91)*DD(17)
where A, B, C, and D represent the time for ultrasonication (min), pH, adsorbent amount (g), and concentration of CV dye (mg L^−1^), respectively. Figure 6 shows the 3D response surface graphs for the current studies.

Figure 7a shows that the percentage of CV removal grew over the ultrasonication time until it reached the optimal timing. Ultrasonic irradiation has increased the likelihood of contact between the adsorbent and adsorbate causing enhanced adsorption [14]. As the concentration of the adsorbate grew, the removal of CV initially increased as well, but it eventually stabilized after the adsorption sites were saturated [31]. Figure 7b shows that the amount of CV that adsorbs to the nanocomposites increases when the pH rises, i.e., pH > pH_PZC_, as a result of the adsorbent developing a negative charge that attracts the cationic organic CV pollutant [50]. Figure 7c shows that the adsorption process improved as the amount of nanocomposites increased, but after achieving the adsorption maximum (the ideal value), the process did not improve any further, even when the amount was increased. This might be due to processes such as agglomeration [51].

### 3.7. Method Validation at the OOP

It is very important to experimentally validate the projected model. By validating the reaction at the recommended OOP, the effectiveness of the proposed approach was confirmed [39]. Nine replicates were carried out at the OOP for validation, and the outcomes are shown in Table 3.

As shown in Table 3, the model is effective at forecasting how well the adsorption process would remove the supplied components. The OOP’s experimental validation yielded a %RSD_9_ of 1.648, which represented the closeness of 9 replicate results values to the predicted values, which shows the significance of the model employed.

### 3.8. Isotherm Analysis

The adsorbate–adsorbent interaction was observed by applying various isotherm models. These isotherm models are crucial in suggesting the adsorption mechanism since they offer an evaluation of the adsorbent’s adsorption efficacy. The adsorption equilibrium was examined in recent works using the Temkin isotherm, Langmuir, Dubinin–Radushkevich, and Freundlich models. The authenticity of these models was evaluated by RMSE, χ^2^, and R^2^ analyses, which are summarized in Table 4.

The Temkin isotherm model, which had the closest value of Q_e_ (mg g^−1^), offered the best fitting, as shown in Table 4, and this is also supported by statistical error assessments. According to the results of the statistical error analysis functions, the Freundlich and D–R isotherm models also offered sufficient fitting. The weak R^2^, χ^2^, and RMSE values revealed that the Langmuir isotherm and Harkins–Jura models were the least fit models. CV dye adsorption on the IZNC is therefore physical in nature that is also affirmed by the D–R adsorption energy that was less than 8 kJ mol^−1^, which is insufficient for the creation of any surface chemical bonds or an ion exchange mechanism [35].

### 3.9. Kinetic Analysis

The adsorption kinetics of CV dye on the IZNC was investigated by the Elovich models, intraparticle diffusion, pseudo-first-order, and pseudo-second-order. The findings of the kinetics models are provided in Table 5.

Due to their outstanding R^2^, χ^2^, and RMSE values, the intraparticle diffusion model and pseudo-second-order kinetic model were noticed to fit on the experimental data. The intraparticle diffusion model had the lowest R^2^ (76.92), χ^2^ (0.06649), and RMSE (1.291), representing that it provided the best match for the experimental data. This demonstrated the presence of distinct pore-filling processes due to the multi-linearity of the intraparticle diffusion plot. Furthermore, a pseudo-second-order kinetic model offered the second-best fit, with χ^2^ (0.1116), which is below the required χ^2^ (0.711) at α = 0.05 and 4th degree of freedom. According to the R^2^ (75.10), χ^2^ (63.65), and RMSE (32.80) values, the Elovich kinetic model offered the third most effective fit to the experimental data. The experimental data were poorly suited to the pseudo-first-order kinetic model, as shown by the R^2^ (61.32), χ^2^ (836.4), and RMSE (369.7) values.

### 3.10. Thermodynamic Studies

The thermodynamic viability and spontaneity of the adsorption process must be evaluated. Below are the formulae used to calculate the thermodynamic parameters.
(18)$$\Delta G^{o}=-RT\ln K$$
(19)$$\ln K_{D}=\frac{\Delta S^{o}}{R}-\frac{\Delta H^{o}}{RT}$$
where T is the temperature (Kelvin), R is the universal gas constant (8.314 Jmol^−1^ K^−1^), ΔS^o^ is the change in entropy, ΔH^o^ is the change in enthalpy, and ΔG^o^ is the change in Gibb’s free energy.

Crystal violet dye adsorption thermodynamic studies on the nanocomposites were performed at 308 K, 313 K, and 318 K. Because of the positive results for the ΔS^o^, especially at high concentrations, and negative results for the change in ΔG^o^, the results show that the adsorption of CV dye on the IZNC was spontaneous. Additionally, it was found that the exothermic nature of the CV dye adsorption on the nanocomposites was evidenced by the negative value of ΔG^o^. The most spontaneous condition of the system was confirmed by thermodynamic analysis to be at moderate temperature, or 313 K, and increased concentration levels, or 20 mg L^−1^ and 25 mg L^−1^, as shown in Table 6 [15].

### 3.11. CV dye Recovery

Adsorbate (CV dye) recovery was performed in acidic, neutral, and basic conditions. It was found that the presence of acetic acid in an acidic environment caused the most desorption.

The pH_PZC_ of the nanocomposites was also discovered to be 6.0. A pH less than 6.0 may cause a complete positive charge on the nanocomposite’s surface, assisting in the desorption of the cationic CV dye in the acidic medium [37]. In addition, the desorption increased linearly (R^2^ = 0.9852) with time in the presence of an acetic acid medium, as demonstrated by the positive slope of the trend lines created on the desorption bar graphs.

The fact that the CH_3_COOH medium showed the steepest slope further supports the acidic medium’s suitability for the desorption procedure of CV dye. Because the surface positive charges were unavailable due to the pH impact, desorption in neutral media was seen to be modestly reduced [37].

Furthermore, the desorption effectiveness of the cationic CV dye and the nanocomposite’s surface was present in an acidic medium, but not in a neutral one. Additionally, the basic medium was examined, and it was found that desorption was minimal. The correlation index is displayed in Figure 8.

### 3.12. Regeneration of Adsorbent

The ultrasonication-assisted desorption process regenerated the surface of the nanocomposites, which allowed the nanocomposites to be used again in the adsorption of the pollutant from the aqueous medium. Figure 9 presents the performance of the regenerated adsorbent for 5 cycles.

The figure reflects that the regenerated adsorbent has tremendous efficacy of decontaminating the aqueous medium from the organic pollutant at an appreciable frequency, and after only 10min of desorption followed by drying, the regenerated adsorbent was ready for another adsorption process. The rapid adsorption–desorption cycle makes it a great contender in dealing with the contemporary water pollution challenges rapidly and effectively.

### 3.13. Interference Ions Effect

To evaluate the adsorption process, the effect of several interference ions was studied at concentrations ranging from 0.1 –0.001 M. These ions were produced by the addition of LiCl, MgCl_2_, KCl, and NaCl. The experimental findings made it evident that the highest adsorption efficacy occurred at a concentration of 0.01 M LiCl matrix ions.

In the case of matrix ions produced by MgCl_2_ and KCl, the maximum adsorption occurred at a low concentration, i.e., 0.001 M. The adsorption efficiency of the process was progressively abated when the concentration was increased to 0.1 M, indicating a decrease in the removal efficiency with respect to an increased concentration of the matrix ions. It is interesting to note that different findings were achieved with NaCl salt because the effectiveness of the adsorption process improved as the concentration increased. The outcomes are displayed in Figure 10.

### 3.14. Operational Cost

When compared to many alternative techniques, such as electrocoagulation, advanced oxidation processes, etc., the overall operational cost was estimated to be 97.90 US dollars per m^3^, or only 9.790 dollars per dm^3^. The breakdown of the costs for removing simulated wastewater contaminant CV dye is presented in Figure 11.

### 3.15. Comparing the Capability of Adsorption with Earlier Work

Nowadays, the adsorption process is used widely. There are many new and inventive adsorbents being formed with unique adsorptive characteristics. New IZNC were created, the same as in the current study, and they demonstrated outstanding CV dye adsorption performance. In Table 7, the comparison of the synthesized nanocomposites’ adsorption effectiveness to the previously reported adsorbents is shown in the bar highlight, as well as in the text column.

## 4. Conclusions

The present study focuses on a facile low-cost green route to synthesize IZNC by *Azadirachta indica* leaves extract. The nanocomposites were characterized for their size, morphology, elemental composition, and the surface functional groups. It was revealed that the nanoparticles are homogeneous in shape with an average size of 100 nm. EDS showed a significant proportion of iron and zinc in the mass percentage of the synthesized nanocomposite. The nanocomposites were used as a high-performance adsorbent for CV dye. The ultrasonically driven adsorption process showed 80.39% dye removal with an adsorption capacity of 40.20 mg g^−1^ under optimized operating parameters (OOP), considering the time of 8.0 min, pH of 9.0, IZNC amount of 25.0 mg, and dye concentration of 25 mg L^−1^. Because a %RSD_9_ (1.648%) was achieved following replication under OOP circumstances, these values have been experimentally validated. According to the best fitted models determined by their R^2^, χ^2^, and RMSE values, Temkin and Freundlich isotherm models fit well into the experimental data, indicating the physical nature of adsorption with a D–R adsorption energy of 2.5 kJmol^−1^. The goodness of kinetic model fitting suggests that the system follows a pseudo-second-order kinetic model with intraparticle diffusion as the prime mechanism for the CV dye adsorption. Thermodynamic analysis proves the spontaneous nature of the adsorption process as defined by negative ΔG^o^, positive ΔS^o^, and negative ΔH^o^ values. An acetic acidic medium was found to be the most practical system among all other media for the recovery of CV dye. The nano-adsorbent showed excellent recycling ability, as significant adsorption was observed after 5 adsorption–desorption cycles. The cationic dye contaminants were easier to desorb due to the surface positive charge caused by pH < pH_PZC_. The adsorption process was quite robust, as the CV dye adsorption was not significantly affected in the presence of several matrix ions produced by salt addition of LiCl, MgCl_2_, KCl, and NaCl. The highest adsorption was shown when LiCl salt was introduced.

The proposed method provides an economical and sustainable approach to synthesize an IZNC for the removal of crystal violet dye. The nanocomposites may be used in the development of a water treatment plant, where they will be applied in a fed bed column followed by a chlorination process for producing pure water for domestic as well as industrial purposes.

## Figures and Tables

**Figure 1 materials-16-01023-f001:**
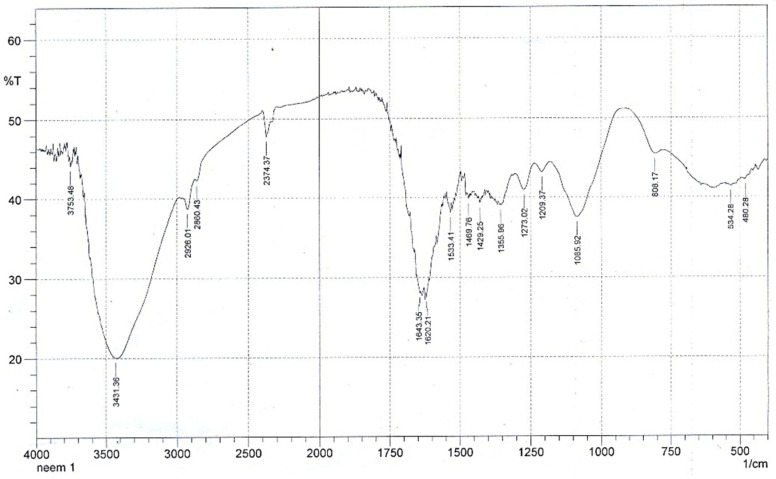
FTIR spectrum of the IZNC.

**Figure 2 materials-16-01023-f002:**
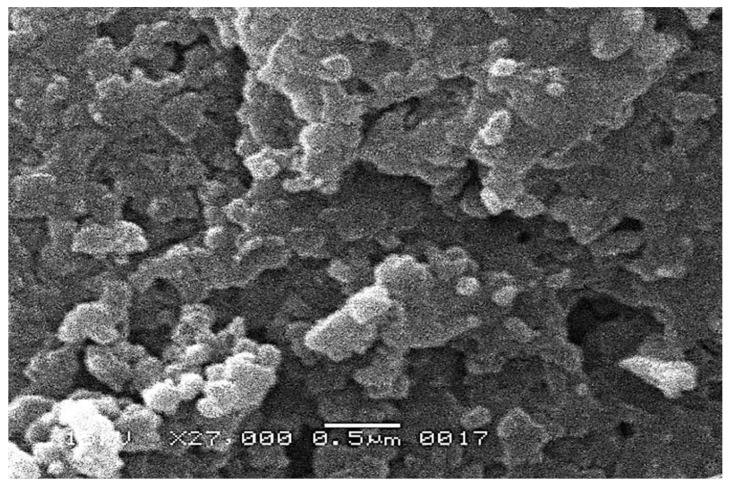
SEM image of the IZNC.

**Figure 3 materials-16-01023-f003:**
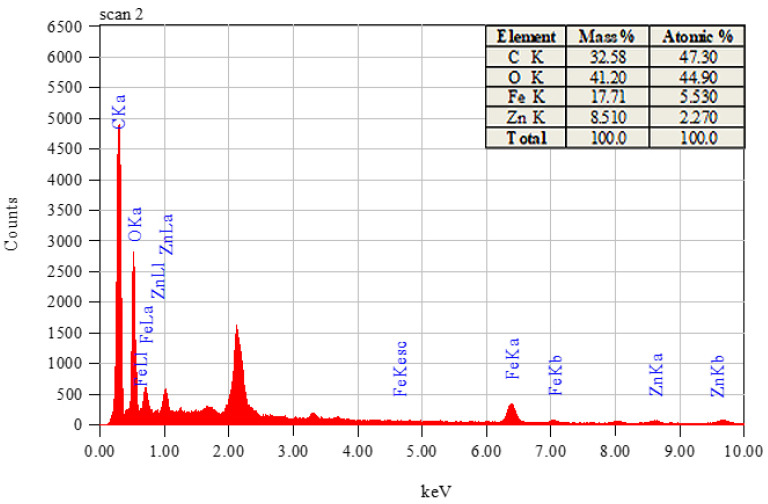
EDS spectra along with elemental composition.

**Figure 4 materials-16-01023-f004:**
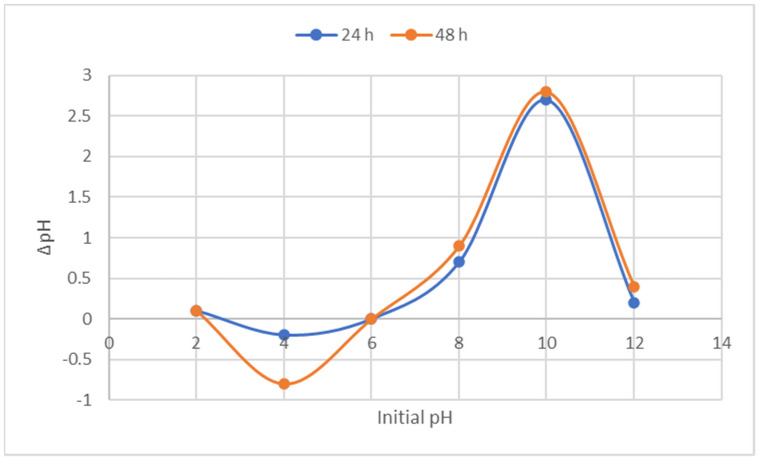
Plot of initial pH versus final pH at 24 h and 48 h.

**Figure 5 materials-16-01023-f005:**
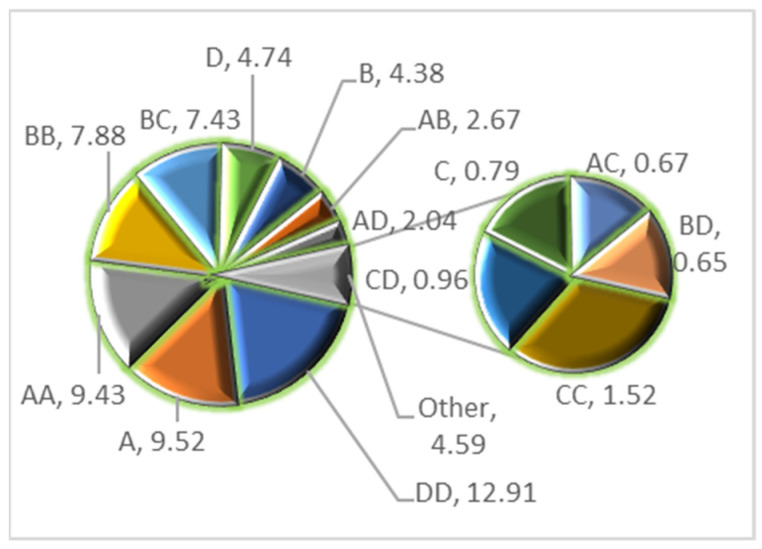
Pareto chart showing the significance of the factors and their combinations.

**Figure 6 materials-16-01023-f006:**
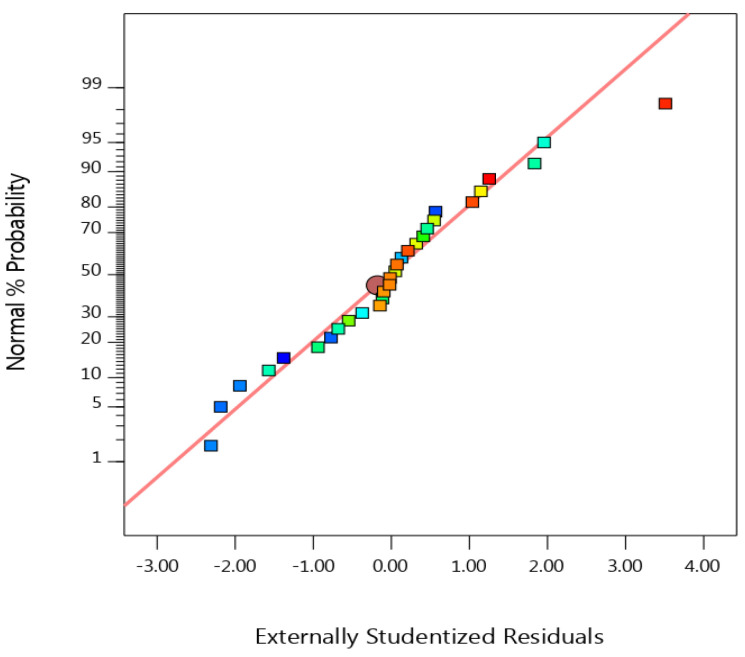
Normal probability plots of residuals.

**Figure 7 materials-16-01023-f007:**
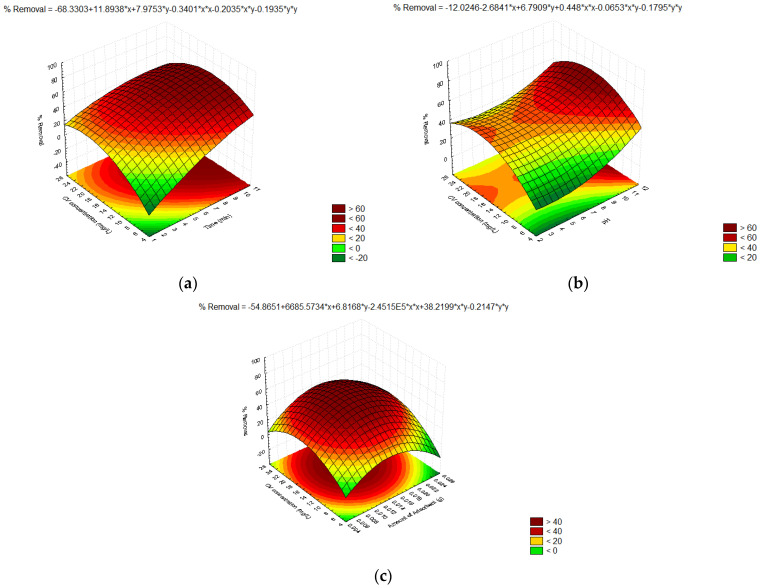
Three-dimensional RSM plots for the adsorptive decontamination of CV by the IZNC as a function of (**a**) time and CV concentration, (**b**) pH and CV concentration, and (**c**) the amount of nanocomposites and CV concentration.

**Figure 8 materials-16-01023-f008:**
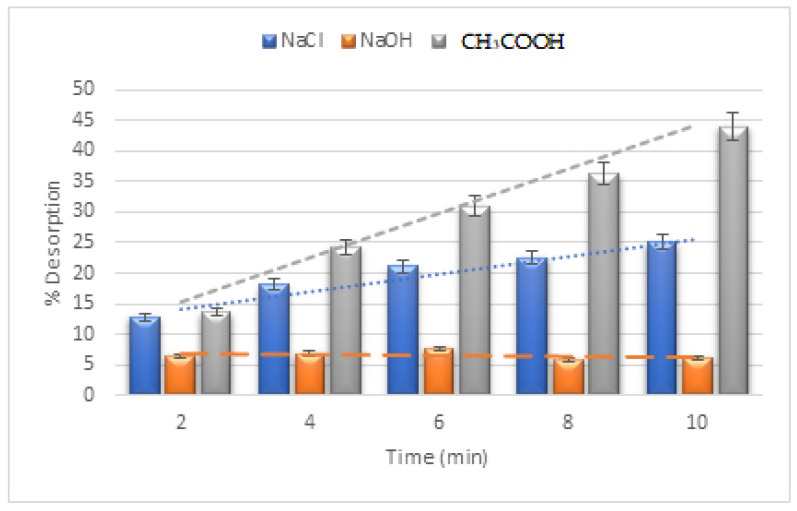
Desorption of CV dye in various media.

**Figure 9 materials-16-01023-f009:**
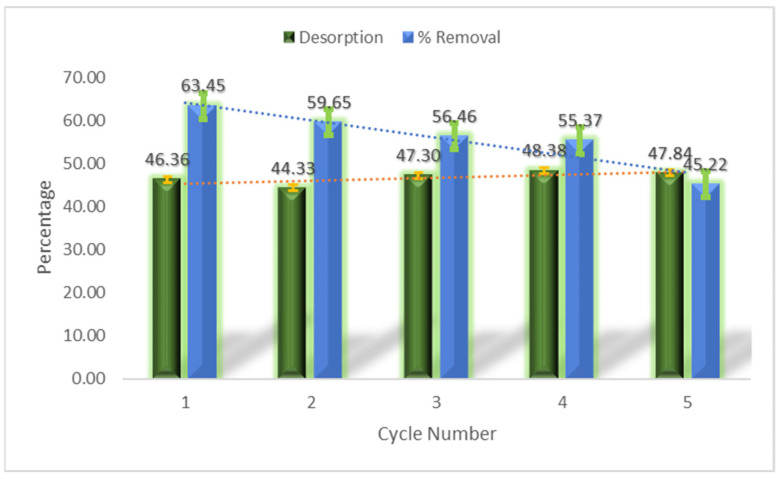
Regeneration of the adsorbent after 5 adsorption–desorption cycles.

**Figure 10 materials-16-01023-f010:**
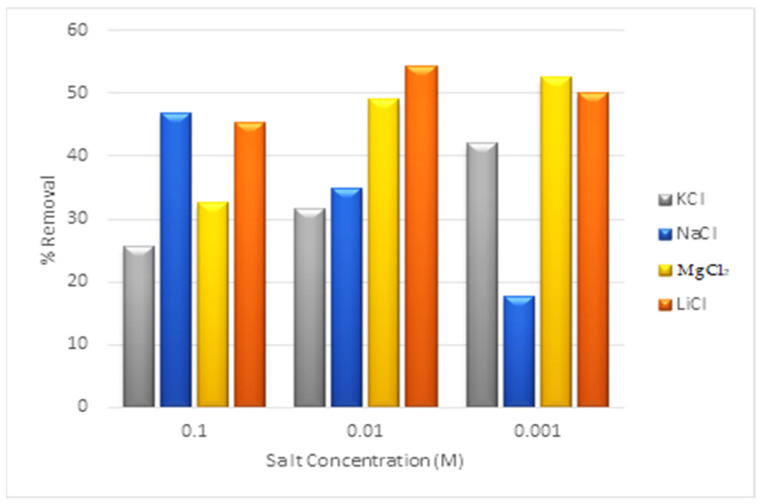
Effect of salts on the efficacy of the adsorption process.

**Figure 11 materials-16-01023-f011:**
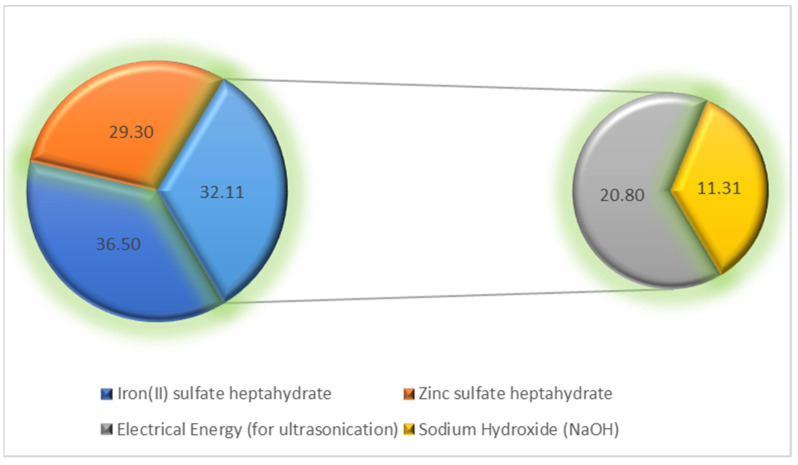
Cost distribution for the removal of CV dye from simulated wastewater at OOP.

**Table 2 materials-16-01023-t002:** ANOVA for the decontamination of CV dye from aqueous medium by the IZNC.

Source	SS	DF	MS	F-Value	*p*-Value
Model	11,300	14	806.8	2.795	0.02872
Time	2175	1	2175	7.536	0.01503
pH	461.3	1	461.3	1.598	0.2254
Amount of adsorbent	15.10	1	15.10	0.0523	0.8222
CV concentration	539.1	1	539.1	1.868	0.1919
AB	114.3	1	114.3	0.3959	0.5387
AC	7.104	1	7.104	0.0246	0.8774
AD	66.26	1	66.26	0.2296	0.6388
BC	882.8	1	882.8	3.059	0.1007
BD	6.818	1	6.818	0.0236	0.8799
CD	14.61	1	14.61	0.0506	0.8250
AA	2438	1	2438	8.448	0.01085
BB	1702	1	1702	5.896	0.02823
CC	63.25	1	63.25	0.219	0.6464
DD	4569	1	4569	15.83	0.00121
Residual	4329	15	288.6	*	*
Lack of Fit	4308	10	430.8	102.6	0.00004
Pure Error	21.00	5	4.201	*	*
Cor Total	15,620	29	*	*	*

* Not available, SS = Sum of square, DF = Degree of freedom, MS = Mean of square.

**Table 3 materials-16-01023-t003:** Optimum operating parameters, model summary, and the validation of the responses at the OOP along with the RSD for the IZNC.

Optimum Parameters		Responses		
		OOP	Replicates’ Responses at OOP	% RSD_9_ at OOP
	Time (min)	8.00	79.72	1.648
	pH	9.00	77.48	
	Amount of adsorbent (mg)	10.0	79.43	
	CV concentration (mg L^−1^)	25.0	78.64	
Predicted response at OOP		80.39	78.85	
%R^2^		72.29	78.58	
%R^2^ Adjusted		46.43	78.10	
CCD Standard Deviation		7.180	75.07	
			78.10	

**Table 4 materials-16-01023-t004:** Analysis of the adsorption of CV dye on the IZNC by various isotherm models.

Model	Parameter	Temperature (K)		
		308	313	318
Langmuir	Q_o_ (mg g^−1^)	−41.23	1483	−1.927
	K_L_ (L g^−1^)	−112.5	7.474	−146.2
	%R^2^	0.3968	0.4634	97.00
	χ^2^	36.01	4.188	39.62
	RMSE	21.07	6.689	27.87
Freundlich	K_F_ (mg g^−1^) (L mg^−1^)^1/n^	2.901	10.75	0.002031
	N	0.5293	0.9758	0.1947
	%R^2^	80.60	92.75	89.71
	χ^2^	21.55	4.316	12.52
	RMSE	16.79	7.139	10.27
Dubinin-Radushkevich	Q_s_ (mg g^−1^)	6.466	10.12	0.0295
	β (mol^2^ kJ^−2^) (×−10^−8^)	10.30	8.256	2.453
	E (kJ mol^−1^)	2.203	2.461	1.428
	%R^2^	75.15	82.10	90.73
	χ^2^	25.50	9.449	11.50
	RMSE	17.29	10.46	9.710
Temkin	A (L g^−1^)	0.4752	1.382	0.2495
	B	77.64	25.43	65.02
	%R^2^	90.25	91.47	61.53
	χ^2^	8.300	2.500	2.293
	RMSE	11.36	5.443	12.54
Harkins–Jura Model	A	76.57	73.49	1.456
	B (×−10^−2^)	75.18	59.66	80.29
	%R^2^	52.12	75.06	95.96
	χ^2^	1666	451.0	18.89
	RMSE	69.28	34.48	12.42

**Table 5 materials-16-01023-t005:** Summary of the kinetics of CV dye adsorption by the IZNC by various kinetic models.

Model	Parameter	Values
Pseudo-First-Order Kinetic Model	k_1_ (min^−1^)	0.2750
	Q_e_ (mg g^−1^)	8.660
	%R^2^	61.32
	χ^2^	836.4
	RMSE	369.7
Pseudo-Second-Order Kinetic Model	k_2_ (g g^−1^) (L mg^−1^)^1/n^	52.67
	Q_e_ (mg g^−1^)	80.89
	%R^2^	99.93
	χ^2^	0.1116
	RMSE	1.661
Intraparticle Diffusion Model	k_id_ (mg g^−1^ min^−0.5^)	3.627
	C (mg g^−1^)	68.59
	%R^2^	76.92
	χ^2^	0.06649
	RMSE	1.291
Elovich Model	a (g g^−1^ min^−1^) (10^4^)	89.75
	b (mg g^−1^) (10^−2^)	27.212
	%R^2^	75.10
	χ^2^	63.65
	RMSE	32.80

**Table 6 materials-16-01023-t006:** Thermodynamic parameters for the adsorption of CV dye on the IZNC.

Concentration (mg L^−1^)	Enthalpy (ΔH) (kJmol^−1^ K^−1^)	Entropy (ΔS) (Jmol^−1^)	ΔG (kJ/mol)		
			308 K	313 K	318 K
5	−65.94	−196.8	−1.475	−2.159	0.1102
10	−37.22	−96.23	−2.645	−2.887	−2.256
15	−22.14	−43.55	−3.092	−3.182	−2.966
20	−0.8586	27.32	−3.249	−3.440	−3.345
25	−1.089	28.59	−3.444	−3.549	−3.546

**Table 7 materials-16-01023-t007:** Comparison of the adsorption efficacy of the IZNC with earlier research.

Adsorbent	CV Adsorption Capacity (mg g^−1^)	References
Fe–Zn nanocomposites	40.20	This research
Magnetic κ-carrageenan-g-poly (methacrylic acid)	28.24	[52]
Coconut husk-based activated carbon	25.74	[51]
TLAC/Chitosan composite	12.50	[53]
Pumice Stone	6.990	[54]
clay/poly(N-isopropylacrylamide) hydrogel	4.710	[21]

## Data Availability

The manuscript is based on novel and original data where it has been determined through thorough experimentation. However, theoretical models and other pieces of information are available online and they have been appropriately reported in the references.

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
