# Peer review of "Sustainable Synthesis of Iron–Zinc Nanocomposites by Azadirachta indica Leaves Extract for RSM-Optimized Sono-Adsorptive Removal of Crystal Violet Dye"

_materials, 2023, doi:10.3390/ma16031023_

Round 1
Reviewer 1 Report
Please find the comments here.
1. It is suggested to provide the individual FTIR for iron oxide and zinc oxide so that comparison can be made among the nanoparticles synthesis. It is encouraged to have the FTIR of the nanocomposites without the addition of AI.
2. Likewise, the SEM result lack of explanation. The picture is not so clear. Also, comparison with the single iron oxide, zinc oxide and fe-zn without AI should be provided to provide a comprehensive outcome on what the AI can alter in the morphology of the composites.
3. Same for EDS, the explanation can be improved. No references are used to support both SEM and EDS section.
4. Reviewer do not understand Figure 4 at all. Instead of providing this figure, the authors should include the charges in Y-axis. What do you mean the curve crossed at pH 6.00? It seems all the curve seems to be at intersect at the same place.
5. Under section 3.2, is there any basis of the selection of these parameters? The basis should be placed under methodology and the selection of these parameters should be supported by references.
6. According to Table 1, the highest removal would be 85%. What happened the removal performance if pure Zinc Oxide and Iron Oxide and composite without AI are used? The study should include this for comparison.
7. Section 3.3, What is A, B, C and D parameter? No label or explanation in the previous section? The explanation should be done in methodology section or previous section prior to Section 3.3 and Section 3.6. This will cause reviewer unable to interpret Figure 4 because of this.
8. What is OOP? Please indicate it.
9. The section 3.8 should be place under methodology while Section 3.8 should focus on the explanation instead. Similar case for Section 3.9.
10. How the intraparticle diffusion model can fit the experimental data with only regression of 76-77%? Usually, this value is not so strong.
11. So what does the Kinetic analysis and isotherm analysis tell the audience about the composite?
12. What is the regeneration of the adsorbent? Only the removal percentage was calculated. How about the regeneration? How many percent of the regeneration? Does the regeneration percentage is adequate to proof the performance of the composites?
13. Element of performance and regeneration of composite should be relate with cost for comprehensive comparison.
14. Is the result of adsorption capacity discuss in the study or shown in the study prior to Table 7?
15. Kindly look at the quality of table, figure, format and some language. All of these are adequate but there is a room for improvement.
Author Response
Thank you very much for your time to review the paper and for your useful and positive comments.
- It is suggested to provide the individual FTIR for iron oxide and zinc oxide so that comparison can be made among the nanoparticles synthesis. It is encouraged to have the FTIR of the nanocomposites without the addition of AI.
Response: Azadirachta Indica (AI) extract was used to synthesize nanoparticles via an eco-friendly approach. Without addition of the extract, the nanoparticles cannot be synthesized.
- Likewise, the SEM result lack of explanation. The picture is not so clear. Also, comparison with the single iron oxide, zinc oxide and fe-zn without AI should be provided to provide a comprehensive outcome on what the AI can alter in the morphology of the composites.
Response: Azadirachta Indica extract was used to synthesize nanoparticles and without its addition, nanoparticles cannot be synthesized and hence, a comparison cannot be made. SEM and EDS explanations have been improved in the text.
- Same for EDS, the explanation can be improved. No references are used to support both SEM and EDS section.
Response: Appropriate references have been placed in the manuscript after SEM and EDS discussions.
- Reviewer do not understand Figure 4 at all. Instead of providing this figure, the authors should include the charges in Y-axis. What do you mean the curve crossed at pH 6.00? It seems all the curve seems to be at intersect at the same place.
Response: Figure 4 has been replaced with slightly modified figure with a little different methodology in which change in pH after 24h and 48h was plotted against initial pH. Reference for the method has also been updated in the text.
- Under section 3.2, is there any basis of the selection of these parameters? The basis should be placed under methodology and the selection of these parameters should be supported by references.
Response: The selection of these parameters was carried out by performing preliminary runs and thorough literature review. Parameters’ range and their rational have been incorporated into the text.
- According to Table 1, the highest removal would be 85%. What happened the removal performance if pure Zinc Oxide and Iron Oxide and composite without AI are used? The study should include this for comparison.
Response: Without the addition of A. Indica leaves extract, nanoparticles cannot be formed. It works as reducing as well as capping agent in the synthesis of nanoparticles. However, few already reported materials for the removal of crystal violet dye from the wastewater have been compared with the synthesized nanomaterial in section 3.14 (Table 7). The synthesized nanoparticles have exceptional adsorption efficiency in comparison to the reported research.
- Section 3.3, What is A, B, C and D parameter? No label or explanation in the previous section? The explanation should be done in methodology section or previous section prior to Section 3.3 and Section 3.6. This will cause reviewer unable to interpret Figure 4 because of this.
Response: A, B, C and D parameters have been discussed in section 3.2 and their ranges have also been explained along with appropriate references.
- What is OOP? Please indicate it.
Response: OOP refers to the ‘Optimum Operating Parameters’ that indicates the set of variables at which the best response is obtained.
- The section 3.8 should be place under methodology while Section 3.8 should focus on the explanation instead. Similar case for Section 3.9.
Response: Section 3.8 and section 3.9 have been placed under methodology.
- How the intraparticle diffusion model can fit the experimental data with only regression of 76-77%? Usually, this value is not so strong.
Response: Intraparticle diffusion model graph gives better R2 when diffusion is the only rate determining step. In presence of more than one mechanism for the adsorption like diffusion and pore filling, the data gets split into two or even more segments of the same line having different slopes giving rise to low R2 values. As in our study, two mechanisms were observed i.e., pore filling mechanism along with diffusion therefore low R2 was observed but the model yielded low χ2, and RMSE values that indicate its fitting to the experimental data.
- So what does the Kinetic analysis and isotherm analysis tell the audience about the composite?
Response: Kinetic analysis tells that how adsorption varies with respect to time and about the mechanism of adsorption. In our case, CV dye adsorption was dependent on pore filling mechanism along with diffusion.
- What is the regeneration of the adsorbent? Only the removal percentage was calculated. How about the regeneration? How many percent of the regeneration? Does the regeneration percentage is adequate to proof the performance of the composites?
Response: Regeneration of adsorbent was ensured by desorbing adsorbate from the exhausted adsorbent. The amount of desorbed adsorbate was calculated as the percentage of adsorbed adsorbate molecule. The regenerated adsorbent was reused in the purification processes and the cycle was repeated for at least 5 times and the results have been shown in Figure 9.
- Element of performance and regeneration of composite should be relate with cost for comprehensive comparison.
Response: Adsorption efficacy was compared with already reported adsorbents that showed immense adsorption potential of the synthesized material. In addition to that, cost analysis was also performed for one time purification process which is reported in Figure 10. Cost analysis data was unavailable in those reported research for comparison.
- Is the result of adsorption capacity discuss in the study or shown in the study prior to Table 7?
Response: The table presents the adsorption capacity with earlier reported adsorbents. The outstanding adsorption capacity has been described in the paragraph prior to table 7. It was necessary to compare the newly synthesized material with earlier reported adsorbents to assess its adsorption performance.
- Kindly look at the quality of table, figure, format and some language. All of these are adequate but there is a room for improvement.
Response: The quality of the images has been optimized.

Reviewer 2 Report
Summary and general comments
In this work, authors synthesized Fe-Zn composite using neem leaf extract and investigated its adsorptive performance in the removal of Crystal violet dye.
The current version of the manuscript can be further improved. Please find the comments below for details.
Specific comments
1. Title: The word: “Azadirachta Indica leaf extract” should be included in the title because the current title is misleading as the current version indicates that the leaf itself is also a part of the adsorbent.
2. Abstract: abbreviations that is used only once in the abstract do not need to be included.
3. Introduction, Following the biology taxonomy’s rule, “Azadirachta Indica” should be mentioned in full and italicized and as “A. Indica” thereafter.
4. The common name of plant A. Indica “neem” should also be mentioned in the introduction.
5. In the introduction, the authors should discuss other prominent choices of adsorbents, such as high-availability sustainable biomass. Here are some of my recommendations: doi.org/10.5004/dwt.2018.21775
doi.org/10.1016/j.jtice.2021.11.001
6. Once an abbreviation is defined, i.e. ZnO, it should be used consistently throughout after its first appearance and avoid switching back to full term
7. All the chemical formulas should be properly formatted. i.e. the numbers were not subscripted.
8. Line 126. What is the original colour, and what colour did the solution change to? These observations should be reported.
9. Section 2.3. The authors should describe the appearance of the synthesized adsorbent. What is the particle size?
10. Section 2.4 is vague. Important details such as the number of scans and resolution for FTIR, and the sputtering detail of SEM should be included. The machine model and manufacturers should also be mentioned.
11. Section 2.6 is not clear. The investigated parameters should be listed. It is not clear how the adsorbent was separated from the mixture after the adsorption process. What are the adsorbate dosage, adsorbate pH, etc? How was the solution shaken?
12. The isotherm and kinetics models should be cited to the work of the original authors. Not Saad et al.
13. Section 3.1 Abbreviations are not required to be defined at every appearance.
14. Fig 2. Authors can choose to show one, as 25000x and 27000x are not much of a big difference,.
15. Recheck section 3.1.4. Usually, (pHf – pHi) vs pHi is plotted. Not pHf vs pHi.
16. The inconsistency of the symbol should be avoided. Qe is used in isotherm modelling and switched to qe in kinetics modelling. Symbols k1 k2 is used in the equations but switched to K1 and K2 in the tables.
17. Is the experiment only carried out for 8min? Has the adsorption even achieved equilibrium? The isotherm equation required values from Ce and Qe where the adsorption equilibrium is achieved. If equilibrium is not achieved, a large error can result from the isotherm modelling. Please show the qe vs t plot with a longer contact time to show that equilibrium can be achieved at 8 min.
18. Unnecessary visual effects should be avoided in all the figures and tables. The shadow should be avoided in all bar graphs. The blue colouration in Table 7 should also be removed
Author Response
Thank you very much for your time to review the paper and for your useful and positive comments.
- Title: The word: “Azadirachta Indica leaf extract” should be included in the title because the current title is misleading as the current version indicates that the leaf itself is also a part of the adsorbent.
Response: Azadirachta Indica leaf extract” has been incorporated in the Title. New title is given below.
Modified Title: Sustainable synthesis of iron-zinc nanocomposite by Azadirachta Indica leaf extract for RSM-optimized sono-adsorptive removal of crystal violet dye
- Abstract: abbreviations that is used only once in the abstract do not need to be included.
Response : The Abbreviations are organized accordingly as DGo, DHo, and DSo
- Introduction, Following the biology taxonomy’s rule, “Azadirachta Indica” should be mentioned in full and italicized and as “A. Indica” thereafter.
Response: They have been mentioned appropriately.
- The common name of plant A. Indica “neem” should also be mentioned in the introduction.
Response: Incorporated in introduction.
- In the introduction, the authors should discuss other prominent choices of adsorbents, such as high-availability sustainable biomass. Here are some of my recommendations: doi.org/10.5004/dwt.2018.21775
doi.org/10.1016/j.jtice.2021.11.001
Response: References are incorporated
doi.org/10.5004/dwt.2018.21775
Zaidi, Nur Afiqah Hazirah Mohamad, et al. "Efficient adsorption of malachite green dye using Artocarpus odoratissimus leaves with artificial neural network modelling." Desalination and Water Treatment 101 (2018): 313-324.
doi.org/10.1016/j.jtice.2021.11.001
Kooh, Muhammad Raziq Rahimi, et al. "Machine learning approaches to predict adsorption capacity of Azolla pinnata in the removal of methylene blue." Journal of the Taiwan Institute of Chemical Engineers 132 (2022): 104134.
- Once an abbreviation is defined, i.e.ZnO, it should be used consistently throughout after its first appearance and avoid switching back to full term
Response: It modified in all content
- All the chemical formulas should be properly formatted. i.e. the numbers were not subscripted.
Response: They have been modified
- Line 126. What is the original colour, and what colour did the solution change to? These observations should be reported.
Response: Reported in section 2.3
In the present study, changes of color from yellow to light brown indicated the formation of ZnO NPs from “A.Indica” leaf extracts .The color from yellowish brown color to black color was observed which indicates the formation of the Fe2O3 nanoparticles. When both of solution of metals mixed and after processing the black colour Zn-Fe-Al composites were obtained.
- Section 2.3. The authors should describe the appearance of the synthesized adsorbent. What is the particle size?
Response: Scanning Electron Microscopy (SEM)
Figure 2 demonstrates that the synthesized nanocomposites had uniform sizes and shapes. The bulk of them were found to be spherical. The average nanoparticle size, as shown in the SEM pictures, was 100 nm.
- Section 2.4 is vague. Important details such as the number of scans and resolution for FTIR, and the sputtering detail of SEM should be included. The machine model and manufacturers should also be mentioned.
Response: SEM instrument of JEOL japan product model# JSM6380A is utilized for Scanning Electron Microscopy (SEM). FTIR instrument having model NICOLET 67000 was operated.
- Section 2.6 is not clear. The investigated parameters should be listed. It is not clear how the adsorbent was separated from the mixture after the adsorption process. What are the adsorbate dosage, adsorbate pH, etc? How was the solution shaken?
Response: Details are shown in Table 1.
- The isotherm and kinetics models should be cited to the work of the original authors. Not Saad et al.
Response: Tahir, Hajira, Muhammad Sultan, and Qazi Jahanzeb. "Removal of basic dye methylene blue by using bioabsorbents Ulva lactuca and Sargassum." African Journal of Biotechnology 7.15 (2008).
Tahir, Hajira, et al. "Application of natural and modified sugar cane bagasse for the removal of dye from aqueous solution." Journal of Saudi Chemical Society 20 (2016): S115-S121.
Tahir, Hajira, et al. "Batch adsorption technique for the removal of malachite green and fast green dyes by using montmorillonite clay as adsorbent." African Journal of Biotechnology 9.48 (2010): 8206-8214.
Tahir, Hajira, Muhammad Sultan, and Zainab Qadir. "Physiochemical modification and characterization of bentonite clay and its application for the removal of reactive dyes." International Journal of Chemistry 5.3 (2013): 19.
- Section 3.1 Abbreviations are not required to be defined at every appearance.
Response: They have been modified.
- Fig 2. Authors can choose to show one, as 25000x and 27000x are not much of a big difference,
Response: One image has been kept with 25000x magnification.
- Recheck section 3.1.4. Usually, (pHf – pHi) vs pHi is plotted. Not pHf vs pHi.
Response: Graph has been redrawn with suggested method. Reference has also been updated pertaining to the described method for evaluating pHPZC.
- The inconsistency of the symbol should be avoided. Qe is used in isotherm modelling and switched to qe in kinetics modelling. Symbols k1 k2 is used in the equations but switched to K1 and K2 in the tables.
Response: Qe has been used for adsorption efficiency at equilibrium throughout the text.
3.8.1. Langmuir Adsorption Isotherms
Where, the equilibrium concentration of the dye remaining in the solution is described by Ce. Langmuir constant is represented by KLwhich desribes the energy , the maximum adsorption efficiency (mg g-1) is indicated by Qo and At equilibrium amount of adsorbate being adsorbed per unit mass of adsorbent is indicated by Qe (mg g-1). The values of constant are calculated from the slope and intercept of the graph plotted between Ce/Qe and 1/Ce .
3.8.2. Freundlich Adsorption Isotherm
Where, the amount of dye adsorbed per unit mass of the adsorbent is described by Qe , the equilibrium concentration of dye is shown by Ce where Kf defines Freundlich constant which expresses adsorption capacity of the adsorbent (mg g-1)and n is the Frundlich constant describes adsorption intensity. This isotherm is additional fasion of langmiur model applicable for multilayer adsorption assume on the reaction on all active sites. This isotherm behaves as a reversible adsorption
3.8.3. Dubinin-Radushkevich Adsorption Isotherm
The amount of adsorbate adsorbed at equilibrium is described by Qe (mg g-1),the theoretical adsorption capacity of the adsorbent is denoted by Qs (mg g-1), Ɛ is D-R constant is indicated by Ɛ which is known as Polanyi potential, the D-R isotherm constant is shown by β (mol2 kJ-2) and the mean free energy of system is described by E (kJ mol-1) . The constant are determined by potting between lnQe Vs Ɛ2. This model assumes as temperature reliant adsorption process
3.8.4. Temkin Adsorption Isotherm
The above equations describes Temkin constant which is indicated by b, which indicates the heat of sorption (J/mol); Temkin isotherm constant is expressed by AT (L/g), R is the gas constant (8.314 J/mol K) and absolute temperature is represented by T(K).
- Is the experiment only carried out for 8min? Has the adsorption even achieved equilibrium? The isotherm equation required values from Ce and Qe where the adsorption equilibrium is achieved. If equilibrium is not achieved, a large error can result from the isotherm modelling. Please show the qe vs t plot with a longer contact time to show that equilibrium can be achieved at 8 min.
Response: Adsorption achieved equilibrium at 8 min as shown by the RSM graphs where adsorption appeared to be ceased at 8 min.
- Unnecessary visual effects should be avoided in all the figures and tables. The shadow should be avoided in all bar graphs. The blue colouration in Table 7 should also be removed
Response: Presentation of the figures in the article has been improved. Blue coloration in Table 7 has been removed.

Author Response
Thank you very much for your time to review the paper and for your useful and positive comments.
- Section 2 I recommend to name Materials and Methods
Response: Thank you and followed. The heading of section 2 is replaced with “Experimental” with “Material and Method”, and the heading of section 2.1 replaced “Material and Method” with “Chemicals”, in the manuscript.
- A space must be placed between numbers and units of measurement.
Response: Thank you. Agreed. Spaces have been added between the numeric value and its unit, throughout the manuscript.
- The title of tables and figures should be given in accordance with the requirements of the journal.
Response: Thank you. Accepted and followed.
- Line 302 (Fe-ZN-Al) need to change (Fe-Zn-Al)
Response: Thank you. Accepted and changed to IZNC.
- In section 3.8-3.9 formulas are not numbered
Response: Thank you. Equation numbers had been added in the manuscript.
- Table 7 efficacy????
Response: The table shows adsorption efficacy of various reported adsorbent with the synthesized nanoadsorbent.
- Table 7- it is not clear why the authors used highlighting
Response: The highlighted bars show the adsorption capacity of each adsorbent graphically. However, as per suggestion of Reviewer 1, the highlights have been removed and only numbers are being used to represent the adsorption efficacy of each adsorbent.
- Author Contributions- Section should be given in accordance with the requirements of the journal.
Response: Thank you. The following paragraph has been added to the manuscript.
Conceptualization, H.T. and M.S. ; methodology, H.T. and S.J.; software, M.S. and W.A.E.; validation, O.A.A.; formal analysis, S.J. and H.T.; investigation, H.T. and K.A.A.; resources, M.S. and S.J.; data curation, H.T. and J.Z.; writing—original draft preparation, H.T., M.S. and J.Z.; writing—review and editing, H.T. and O.A.A.; visualization, W.A.E. and K.A.A.; supervision, H.T.; project administration M.S.; funding acquisition, O.A.A., W.A.E. and K.A.A. All authors have read and agreed to the published version of the manuscript.
- References- Section should be given in accordance with the requirements of the journal.
Response: References have been modified and given in the journal according to the accepted style by EndNote 9.

Round 2
Reviewer 2 Report
After the revision, the overall quality of the manuscript has improved